# The Divergent Effects of Ovarian Steroid Hormones in the MCF-7 Model for Luminal A Breast Cancer: Mechanistic Leads for Therapy

**DOI:** 10.3390/ijms23094800

**Published:** 2022-04-27

**Authors:** Nitin T. Telang

**Affiliations:** Cancer Prevention Research Program, Palindrome Liaisons Consultants, Montvale, NJ 07645-1559, USA; ntelang3@gmail.com

**Keywords:** breast carcinoma, steroid hormone metabolism, cancer stem cells, natural products

## Abstract

The growth modulating effects of the ovarian steroid hormones 17β-estradiol (E_2_) and progesterone (PRG) on endocrine-responsive target tissues are well established. In hormone-receptor-positive breast cancer, E_2_ functions as a potent growth promoter, while the function of PRG is less defined. In the hormone-receptor-positive Luminal A and Luminal B molecular subtypes of clinical breast cancer, conventional endocrine therapy predominantly targets estrogen receptor function and estrogen biosynthesis and/or growth factor receptors. These therapeutic options are associated with systemic toxicity, acquired tumor resistance, and the emergence of drug-resistant cancer stem cells, facilitating the progression of therapy-resistant disease. The limitations of targeted endocrine therapy emphasize the identification of nontoxic testable alternatives. In the human breast, carcinoma-derived hormone-receptor-positive MCF-7 model treatment with E_2_ within the physiological concentration range of 1 nM to 20 nM induces progressive growth, upregulated cell cycle progression, and downregulated cellular apoptosis. In contrast, treatment with PRG at the equimolar concentration range exhibits dose-dependent growth inhibition, downregulated cell-cycle progression, and upregulated cellular apoptosis. Nontoxic nutritional herbs at their respective maximum cytostatic concentrations (IC_90_) effectively increase the E_2_ metabolite ratio in favor of the anti-proliferative metabolite. The long-term exposure to the selective estrogen-receptor modulator tamoxifen selects a drug-resistant phenotype, exhibiting increased expressions of stem cell markers. The present review discusses the published evidence relevant to hormone metabolism, growth modulation by hormone metabolites, drug-resistant stem cells, and growth-inhibitory efficacy of nutritional herbs. Collectively, this evidence provides proof of the concept for future research directions that are focused on novel therapeutic options for endocrine therapy-resistant breast cancer that may operate via E_2_- and/or PRG-mediated growth regulation.

## 1. Introduction

In the development of breast cancer, the ovarian steroid hormones 17β-estradiol (E_2_) and progesterone (PRG) represent important growth-regulatory hormones [1]. Although the growth-promoting effects of E_2_ are well-established, the cellular effects of PRG are divergent and mostly context dependent. The receptors for both these hormones function as ligand-regulated nuclear transcription factors that bind to specific DNA response elements and operate via co-activators and co-repressors to affect the transcriptional modulation of down-stream target genes. The critical molecular processes responsible for the cellular effects of E_2_ and PRG include complex autocrine and paracrine pathways involving specific cell-cycle regulators and NFkB-mediated growth modulation. Thus, two steroid hormones effectively interact via well-defined molecular cross-talk to regulate cell proliferation, differentiation, and breast carcinogenesis [2,3,4]. The molecular interaction of estrogen receptor (ER) and progesterone receptor (PR) signaling in hormone-receptor-positive breast cancer results in a modulatory effect of PR on the action of ER to attenuate tumor growth via multiple pathways. In addition to ER- and PR-mediated signaling, the receptor-independent cellular metabolism of these hormones generates metabolites with distinct cellular effects relevant to the process of carcinogenesis [5,6,7,8].

The hormone-receptor-positive Luminal A and Luminal B molecular subtypes of clinical breast cancer respond to selective estrogen-receptor modulators, aromatase inhibitors, and to HER-2 targeted therapeutics. However, long-term anti-estrogen therapy and human epidermal growth factor-2 (HER-2)-based targeted therapy is associated with intrinsic and/or acquired drug resistance that compromises the therapeutic efficacy and favors progression of therapy-resistant disease. The progression of therapy-resistant disease is frequently associated with the emergence of drug-resistant stem cell populations [9]. These limitations of current therapy for breast cancer emphasize an unmet need to identify stem cell targeting testable alternatives. Drug-resistant stem cell models have been developed for HER-2-enriched and triple-negative breast cancer subtypes [10,11].

The expression status of select hormone and growth-factor receptors has provided a basis for the clinically relevant classification of the cellular models for breast cancer subtypes [12,13]. Furthermore, stable expressions of the clinically relevant HER-2 oncogene [14,15,16] or the gene for aromatase enzyme [17,18] have provided valuable models to examine the role of oncogenes and estrogens in breast cancer.

The human mammary carcinoma-derived MCF-7 cell line that is ER/PR positive and expresses non-amplified HER-2, represents a model for the Luminal A molecular subtype of clinical breast cancer [12,13]. Several mechanistically distinct Chinese nutritional herbs have documented preferential growth-inhibitory efficacy in the isogenic MCF-7 phenotypes that exhibit a modulated ER-α function [19]. However, little evidence is available on the effects of nutritional herbs, either on PRG activity or function. Based on the evidence of negative growth regulation by PRG, the potential therapeutic utility of progestogens and progesterone receptor (PR) agonists may represent beneficial treatment options. It is conceivable that the non-toxic nutritional herbs may also be effective in the hormone-receptor-positive breast cancer via PR function.

The goal of the present review is to provide (i) a systematic discussion of the published literature relevant to the development and characterization of the MCF-7 model; (ii) mechanistic leads for the significance of anti-proliferative metabolites of E_2_ and PRG; and (iii) growth-inhibitory efficacy of dietary phytochemicals and nutritional herbs as stem-cell-targeting testable alternatives for the chemo-endocrine therapy-resistant Luminal A molecular subtype of breast cancer.

## 2. Cellular Models

In the mouse mammary gland organ culture model, the ovarian steroid hormones E_2_ and PRG are critical for the induction of normal ductal morphogenesis. Lactogenic hormones, prolactin and hydrocortisone, induce lobulo-alveolar growth and mammary-specific differentiation [20]. In response to treatment with chemical carcinogens, the mammary gland organ cultures develop lactogenic hormone-independent pre-neoplastic alveolar lesions [21], and the transplantation of cells from 7–12 dimethyl benzanthracene (DMBA)-induced alveolar lesions produce rapidly growing tumors [22]. However, the impact of similar interactions between steroid and polypeptide hormones on the initiation or progression of human mammary carcinogenesis remains to be fully elucidated.

Global gene-expression profiling of clinical breast cancer has facilitated the molecular classification of breast cancer subtypes [23]. Cellular models developed from human breast carcinoma-derived cell lines continue to represent valuable resources to identify clinically relevant mechanistic pathways and molecular targets for therapeutic efficacy [12,13].

MCF-7 cells expressing mutant HER-2 represent an additional model for the Luminal B breast cancer subtype [14]. The CYP19 A1 aromatase enzyme is critical for peripheral and intra-tumoral estrogen biosynthesis. This enzyme converts adrenal androstenedione to estrone (E_1_) and testosterone to E_2_. E_1_ is converted to E_2_ by 17β-hydroxysteroid dehydrogenase [5,6]. MCF-7 cells expressing the aromatase gene represent a model for aromatase-positive post-menopausal breast cancer [17,18]. Furthermore, the stable expression of the HER-2 oncogene in human mammary epithelial 184-B5 cells induces tumorigenic transformation [15,16]. The 184-B5/HER cell line represents a valuable model to examine the role of the HER-2 oncogene in the initiation and progression of human breast cancer.

Collectively, the molecular characteristics of various cellular models provide relevant quantitative end-point parameters for the preventive/therapeutic efficacy of dietary phytochemicals and nutritional herbs directly on the target cells of breast cancer. The data on the characteristics of clinically relevant cellular models are summarized in Table 1.

### 2.1. The Growth Characteristics of the MCF-7 Model

Carcinoma-derived cell lines commonly exhibit hyper-proliferation and persistence anchorage-independent growth in vitro, and tumor formation in vivo. Table 2 compares the growth pattern of human breast epithelium-derived non-tumorigenic 184-B5 cells and human breast carcinoma-derived tumorigenic MCF-7 cells. The MCF-7 cells exhibit hyper-proliferation, as evidenced by a decrease in the population doubling time, increase in the saturation density, and accelerated cell cycle progression. Additionally, MCF-7 cells exhibit downregulated cellular apoptosis, decreased estrogen metabolite ratio, and a robust increase in anchorage-independent colony formation, the latter being a specific in vitro surrogate end point for in vivo tumor formation. These data suggest that MCF-7 cells exhibit a loss of homeostatic growth control and persistent cancer risk.

### 2.2. Growth Modulation by Estradiol and Progesterone

The distinct growth modulatory effects of E_2_ and PRG at the physiologically relevant concentrations were examined on MCF-7 cells maintained in a culture medium supplemented by 0.7% serum (serum concentration of E_2_ and PRG < 0.01 nM in the culture medium). Treatment with E_2_ at the physiological concentration range of 1 nM to 20 nM, resulted in a concentration-dependent increase in the viable cell number (Figure 1A). In contrast, treatment with PRG at the equimolar concentration range displayed a concentration-dependent decrease in the viable cell number (Figure 1B).

### 2.3. Cell Cycle Progression and Cellular Apoptosis

The data provided in Figure 2 provide evidence for the potential mechanistic leads that are responsible for the effects of E_2_ and PRG. As illustrated in Figure 2A, treatment with E_2_ decreases the G_1_:S + G_2_/M ratio due to an increase in the S phase of the cell cycle. In contrast, treatment with PRG increases the G_1_:S + G_2_/M ratio due to G_1_ arrest and a decrease in the S and G_2_/M phases of the cell cycle. As illustrated in Figure 2B, treatment with E_2_ inhibits cellular apoptosis, while treatment with PRG increases cellular apoptosis.

ER and PR belong to a superfamily of ligand-regulated nuclear transcription factors that are responsible for the expression of cognate downstream target genes. These genes include pS2, GRB2, and cyclin D1 for E_2_ [3,24], and a receptor activator of nuclear factor kB (RANK) and its ligand RANK-L for PRG [2,3,4]. In addition to the genomic mechanisms, the cellular metabolism of E_2_ and PRG plays a significant modulatory role in breast carcinogenesis. For example, the metabolites generated from E_2_ and PRG have documented divergent growth-modulatory effects on breast cancer cells.

## 3. Hormone Metabolism

### 3.1. Cellular Metabolism of Estradiol

The CYP450-mediated enzymatic metabolism of E_2_ generates mechanistically distinct metabolites that exert specific biological effects on the non-tumorigenic or tumorigenic mammary epithelial cells. During E_2_ metabolism, 4-hydroxylated E_2_ and 16α-hydroxylated E_1_ function as proliferative agents, while 2-hydroxylated E_2_ and 2-hydroxyalted E_1_ function as anti-proliferative agents [5,6]. These metabolites, because of their distinct biological effects, alter the ratio of anti-proliferative to proliferative metabolites. Similar metabolic alterations have also been documented in mammary epithelial cells that exhibit a tumorigenic transformation induced by stable transfection with Ras, Myc, and HER-2 oncogenes [25,26,27]. Thus, the altered 2-OHE_1_:16α-OHE_1_ ratio may represent a novel experimentally modifiable endocrine biomarker for the efficacy of testable alternatives for breast cancer prevention/therapy.

The published data supports the concept that genotoxic E_2_ metabolites 4-hydroxy estradiol (4-OHE_2_), 2-hydroxy estradiol (2-OHE_2_), and 16α-hydroxy estradiol (16α-OHE_2_) induce neoplastic transformation in the non-tumorigenic human mammary epithelial MCF-10F model [5]. In the hormone responsive MCF-7 model, E_2_ has been documented to generate genotoxic adenine and guanine DNA adducts, leading to error-prone DNA repair and/or DNA mutations; and, in the hormone responsive T47D model, 16α-hydroxylated metabolite and 2-hyrdroxylated metabolite of E_2_ exhibit estrogenic and anti-estrogenic activities, respectively [6,28]. Furthermore, it is also notable that the E_2_ metabolite 16α-OHE_1_ induces DNA damage and repair and transformation in non-tumorigenic C57MG cells, while 2-OHE_1_ fails to induce these genotoxic effects [29]. In the MCF-7 model, 16α-OHE1 enhances, while 2-OHE_1_ inhibits in vivo tumor growth [30]. Thus, in addition to ER-α-dependent growth-promoting effects on breast cancer cells, genotoxic E_2_ metabolites may function as initiators of carcinogenesis.

Because of the distinct effects of hydroxylated metabolites of E_1_, a ratio of these metabolites may represent a valuable end-point marker. Table 3 compares the status of the 2-OHE_1_:16α-OHE_1_ ratio in cellular models for breast cancer that differ in their relative risk for cancer development. These data demonstrate that, depending on the relative risk of developing cancer, the estrogen metabolite ratios are substantially decreased.

### 3.2. The Cellular Metabolism of Progesterone

During PRG metabolism, 5α-hydroxylated metabolite functions as a proliferative agent, while 3α-hydroxylated metabolite functions as an anti-proliferative agent [7,8]. Thus, in the tumorigenic cells, the ratio of anti-proliferative:proliferative metabolites of PRG is altered in favor of the proliferative metabolites.

Similar to the E_2_ metabolites, PRG metabolites 5α-dihydro progesterone (5α-PRG) and 3α-dihydro progesterone (3α-PRG) have documented growth-modulatory divergent effects in the MCF-7, MDA-MB-231, T47D, and MCF-10A models. At the mechanistic levels, growth promotion by the 5α metabolite is associated with increased DNA synthesis and the mitotic index and the downregulation of apoptosis, while growth inhibition by the 3α metabolite is associated with a decrease in these end points [7,31,32]. The growth modulating effects of PRG metabolites have also been documented in vivo in the tumors produced by transplanted MDA-MB-231 cells. [8].

The divergent effects of E_2_ and PRG have been documented in T47D, ZR75-1, and MCF-7 cells. For example, the E_2_-mediated inhibition of cellular apoptosis is associated with the increased expression of anti-apoptotic BCL-2. In contrast, the PRG-mediated induction of cellular apoptosis is associated with the decreased BCL-2: BAX ratio, predominantly due to the increased expression of pro-apoptotic BAX [7]. In tissue explants of ER-positive tumors, E_2_ increases, while PRG decreases, cell proliferative activity [32,33].

The autocrine growth-modulatory effects of PRG involve the expression of cyclin-dependent kinase inhibitors p18 and p27 [34], and the modulation of ER-α and RNA polymerase III [35]. In addition to these autocrine effects, the paracrine cellular effects of PRG involve the NFkB pathway via modulation in the expression of RANK and its ligand RANKL [36], and the downregulated select interferon-stimulated genes [37]. Since E_2_ metabolites 4-OHE_2_ and 2-OHE_2_, E_1_ metabolites 16α-OHE_1_ and 2-OHE_1_, and PRG metabolites 5α-PRG and 3α-PRG exhibit opposing growth-regulatory effects, molecular/metabolic pathways for E_2_ and the PRG function may provide potential mechanistic leads responsible for E_2_- and PRG-mediated growth modulation.

Collectively, the evidence for the enzymatic conversion of E_2_ and PRG that generate metabolites with divergent growth-modulatory effects supports the significance of the ratio of proliferative and anti-proliferative metabolites. Thus, the experimental upregulation of anti-proliferative metabolites may provide mechanistic leads for novel therapeutic interventions.

## 4. Efficacy of Natural Products

Dietary phytochemicals, including gluco-brassinins, polyphenols, isoflavones, and terpepnoids induce cell cycle arrest via G_1_ phase arrest, inhibit pHER-2 expression, induce cellular apoptosis, modulate the expressions of apoptosis-specific BCL-2 and BAX, and alter the cellular metabolism of E_2_ in cellular models for breast cancer [38,39,40,41,42]. Mechanistically distinct Chinese nutritional herbs *Cornus officinalis* (CO), *Psoralea corylifolia* (PC), and *Dipsacus apsperoides* (DA) function as effective growth-inhibitory agents in MDA-MB-231 cells, a cellular model for triple-negative breast cancer [42,43,44,45].

The published data on inhibitory efficacy Chinese nutritional herbs *Epimedium grandiflorum* (EG), *Lycium barbarum* (LB), and *Cornus officinalis* (CO) on MCF-7 cells, a model for Luminal A breast cancer, provide mechanistic leads to identify nontoxic testable alternatives. The anti-proliferative effects of these herbs are associated with the altered cellular metabolism of E_2_ in favor of the formation of the non-proliferative metabolite 2-OHE_1_ [46,47,48].

The data provided in Table 4 illustrates that treatment with CO, EG, and LB induces the upregulation of the 2-hydroxylation pathway, leading to an increase in 2-OHE_1_ formation and the downregulation of the 16α-hydroxylation pathway leading to a decrease in the 16α-OHE_1_ formation.

The CYP19 A1 Aromatase enzyme represents a critical enzyme for peripheral and intra-tumoral estrogen biosynthesis in post-menopausal breast cancer, and thereby provides growth-promoting estrogens via the conversion of E_2_ and E_1_ from testosterone and androstenedione, respectively [49,50]. In this context, it is notable that the progesterone metabolite 20α-DHP, predominantly detected in normal breasts, functions as a potent inhibitor of aromatase [51]. An extract from the inner bark of the South American *Tabebuia avellanedae* (TA) tree displays pro-apoptotic and anti-aromatase activity in the MCF-7 ^AROM^ model for aromatase expressing post-menopausal breast cancer. In this model, the TA-mediated induction of cellular apoptosis is associated with the upregulated expression of the pro-apoptotic BAX gene and downregulated expression of the anti-apoptotic BCL-2 gene. Treatment with TA is also associated with downregulated expressions of several E2 responsive genes, such as ESR-1, AORM, PR, PS2, GRB2, and cyclin D1. The potency of aromatase inhibition by TA, based on the content of the active agent naphthofuran dione, is substantially higher than the pharmacological inhibitors of aromatase, letrozole, and exemestane [52]. Collectively, these data on aromatase inhibition provide evidence for aromatase as a potential target for experimental modulation.

## 5. Drug-Resistant Stem Cells

Stem cells are responsible for preserving the regulated program of epithelial proliferation, differentiation, and apoptosis, which are critical for cellular homeostasis in normal breasts via Wnt/β-catenin, Notch, and NFkB signaling pathways, wherein the interactive influence of E_2_ and PRG plays an important role [53]. In drug-resistant cancer stem cells, these signaling pathways are dysregulated, and RAS-, PI3K-, AKT-, and mTOR-mediated survival pathways are activated [9,54,55].

Acquired resistance to conventional chemo-endocrine therapy and molecularly targeted therapy results in the emergence of drug-resistant cancer stem cells. Reliable stem cell models provide valuable experimental approaches to identify stem cell targeting testable alternatives. The tamoxifen-resistant (TAM-R) stem cell model for Luminal A breast cancer, the lapatinib-resistant (LAP-R) stem cell model for HER-2-enriched breast cancer, and the doxorubicin-resistant (DOX-R) stem cell model for triple-negative breast cancer exhibit upregulated expressions of select stem cell markers [10]. These models may provide clinically relevant experimental approaches to evaluate the therapeutic efficacy of stem-cell-targeting natural herbal products, including dietary phytochemicals and Chinese nutritional herbs.

In the Luminal A molecular subtype of breast cancer selective estrogen-receptor modulator tamoxifen (TAM) has wide clinical applications. However, long-term treatment with TAM is associated with acquired tumor resistance. The data in Figure 3 illustrate that the tamoxifen-resistant stem cell model derived from MCF-7 cells exhibits an increased expression of select stem cell markers, such as TS, CD44, NANOG, and OCT-4.

This model may provide clinically relevant experimental approaches to evaluate the therapeutic efficacy of natural herbal products targeted towards breast cancer stem cells.

## 6. Stem Cell Targeting Agents

In the LAP-R model for HER-2-enriched breast cancer, vitamin A derivative all-trans retinoic acid (ATRA) and a natural terpenoid carnosol (CSOL) downregulate stem cell markers [10]. In a model for TNBC, sulforaphane documented stem-cell-selective inhibitory efficacy [56] and benzyl isothio cyanate inhibits mammary stem-like cells functioning via the Klf-4/p21 ^CIP1^ axis [57]. Additionally, Chinese medicines and their constitutive active components [58], dietary phytochemicals [59], and natural products [60,61] have been considered as potential stem-cell-targeting agents.

## 7. Conclusions

The present review discussed the published evidence relevant to the roles of E_2_ and PRG in breast cancer biology, the significance of reliable stem cell models for therapy-resistant breast cancer, and the mechanistic evidence for the growth-inhibitory efficacy of natural products, including dietary phytochemicals and nutritional herbs. Collectively, this review provides a proof concept that natural products may represent testable alternatives for therapy-resistant breast cancer.

A comprehensive overview of the conceptual background of the cellular models for Luminal A, Luminal B, and aromatase-expressing post-menopausal breast cancer subtypes, current targeted therapeutic options, therapy-resistant stem cells, naturally occurring dietary phytochemicals, and nutritional herbs as therapeutic alternatives for therapy-resistant breast cancer and future research directions are summarized in Table 5.

## 8. Future Prospects

By extending the evidence provided in the present review, future research directions will involve patient-derived tumor xenograft (PDTX) and organoid (PDTO) models [62,63,64,65,66]. Additionally, future investigations will develop reliable stem cell models from therapy-resistant breast cancer subtypes. The outcome of these investigations on patient-derived samples is expected to provide strong evidence for clinically relevant data and their potential for clinical translation.

## Figures and Tables

**Figure 1 ijms-23-04800-f001:**
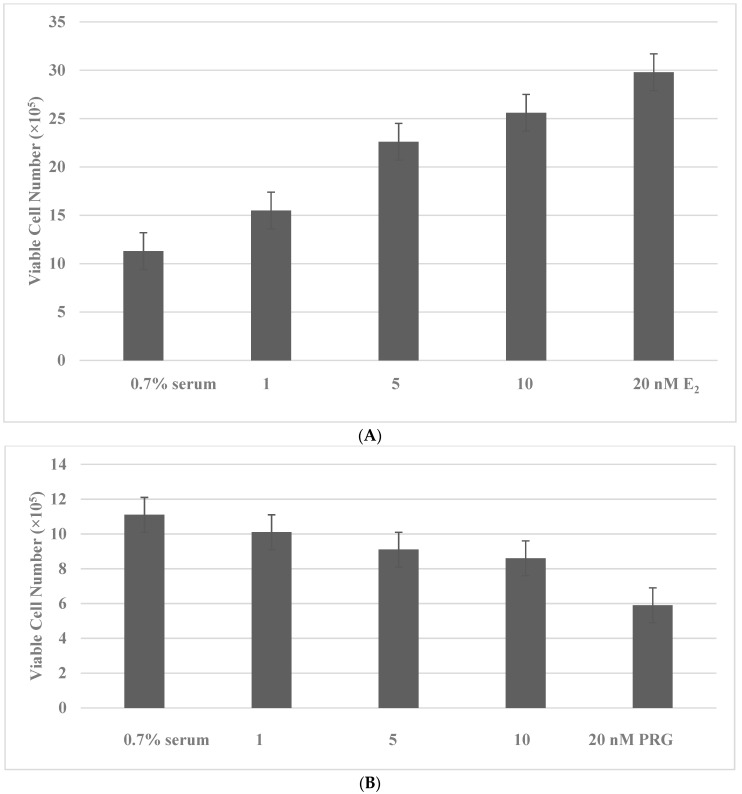
Growth modulatory effects of E_2_ and PRG. (**A**): Treatment with E_2_ exhibits a dose-dependent increase in the viable cell number. (**B**): Treatment with PRG exhibits a dose-dependent decrease in the viable cell number. E_2_, 17β-estradiol; PRG, progesterone.

**Figure 2 ijms-23-04800-f002:**
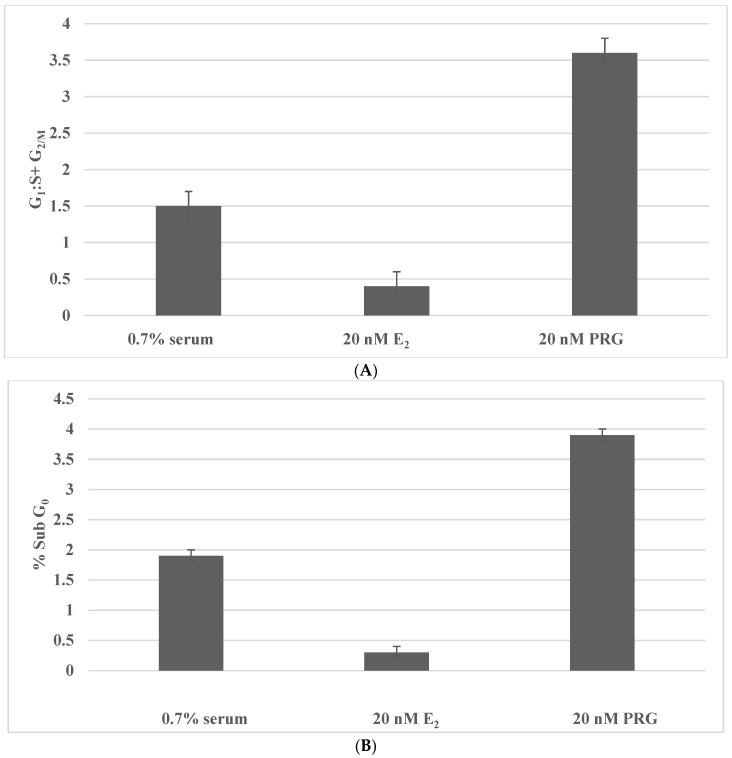
Effects of E_2_ and PRG on cell cycle progression and cellular apoptosis. (**A**): Treatment with 20 nM E_2_ inhibits the G_1_:S + G_2_/M ratio. Treatment with 20 nM PRG increases the G_1_:S + G_2_/M ratio. (**B**): Treatment with 20 nM E2 inhibits % SubG_0_ apoptotic cell population. Treatment with 20 nM PRG induces % Sub G_0_ apoptotic cell population. E2, 17β-estradiol; PRG, progesterone.

**Figure 3 ijms-23-04800-f003:**
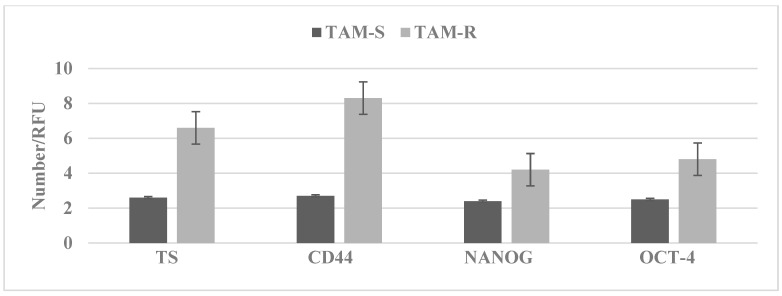
Stem cell marker expression in TAM-R cells. TAM-R cells exhibit increased expressions of TS, CD44, NANOG, and OCT-4 relative to the TAM-S cells. TAM-R, tamoxifen resistant; TAM-S, tamoxifen sensitive; TS, tumor spheroid number; CD44, cluster of differentiation; NANOG, DNA-binding transcription factor; OCT-4, octamer-binding transcription factor-4; and RFU, relative fluorescent unit.

**Table 1 ijms-23-04800-t001:** Cellular models for breast cancer subtypes.

Model	Receptor Status			Subtype	References
	ER	PR	HER-2		
MCF-7	+	+	−	Luminal A	[12,13]
T47D	+	+	−	Luminal A	[12,13]
BT474	+	+	+	Luminal B	[12,13]
MDA-MB-361	+	+	+	Luminal B	[12,13]
MCF-7 HER	+	+	+	Luminal B	[14]
MCF-7AROM	+	+	−	Aromatase positive	[15,16]
SKBr-3	−	−	+	HER-2 enriched	[12,13]
184-B5/HER	−	−	+	HER-2 enriched	[17,18]
MDA-MB-231	−	−	−	Triple negative	[12,13]

ER, estrogen receptor-α; PR, progesterone receptor; and HER-2, human epidermal growth-factor receptor-2.

**Table 2 ijms-23-04800-t002:** Growth pattern of the Luminal A MCF-7 model.

End-Point Biomarker	Experimental Model		
	184-B5	MCF-7	Relative to 184-B5
Population doubling (h)	34.0 ± 1.8	15.2 ± 0.9	−55.3%
Saturation density (×10^5^)	22.3 ± 1.2	26.6 ± 1.7	+19.3%
G_1_:S + G_2_/M ratio	2.3 ± 0.3	1.4 ± 0.4	−39.1%
Sub G_0_ population (%)	14.8 ± 2.3	2.8 ± 1.4	−81.1%
2-OHE_1_:16α-OHE_1_ ratio	6.4 ± 0.8	0.4 ± 0.2	−93.7%
AI colonies	0/18	18/18	+100%

184-B5, non-tumorigenic breast epithelial cells; MCF-7, tumorigenic breast carcinoma cells; 2-OHE_1_; 2-hydrroxyestrone; 16α-OHE_1_, 16α-hydroxyestrone; and AI, anchorage independent.

**Table 3 ijms-23-04800-t003:** Status of the estrogen metabolite ratio.

Model	Relative Cancer Risk	2-OHE_1_:16α-OHE_1_ Ratio	Relative to Low Risk
TDLU reduction mammoplasty	Low	4.8 ± 0.6	-
TDLU breast cancer	High	0.3 ± 0.1	−93.7%
184-B5 breast epithelial cells	Low	6.4 ± 0.8	-
184-B5/HER HER-2 positive	High	0.6 ± 0.3	−90.6%
MCF-7 breast carcinoma	High	0.3 ± 0.1	−95.3%
MMEC mouse mammary epithelial Cells	Low	2.2 ± 0.3	-
MMEC-Ras Ras positive	High	0.2 ± 0.1	−90.9%
MMEC-Myc Myc positive	High	0.3 ± 0.1	−86.4%

TDLU, terminal duct lobular unit; HER-2, human epidermal growth-factor receptor-2; 2-OHE_1_, 2-hydroxyestrone; and 16α-OHE_1_, 16α-hydroxyestrone.

**Table 4 ijms-23-04800-t004:** Altered metabolism of 17β-estradiol by nutritional herbs.

Treatment	Source	Concentration	2-OHE_1_:16α-OHE_1_ Ratio	Relative to E_2_ Control
E_2_ control		20 nM	0.4 ± 0.2	-
E_2_ + EG	Leaf/stem	20 nM + 9.0 µg/mL	1.9 ± 0.2	+3.7x
E_2_ + LB	Bark	20 nM + 0.5 µg/mL	5.2 ± 0.7	+12.0x
E_2_ + CO	Fruit	20 nM + 5.0 µg/mL	6.8 ± 0.8	+16.0x

2-OHE_1_, 2-hydroxyestrone; 16α-OHE_1_, 16α-hydroxyestrone; E_2_, 17β-estradiol; EG, Epimedium grandiflorum; LB, Lycium barbarum; and CO, Cornus officinalis.

**Table 5 ijms-23-04800-t005:** Overview: Luminal breast cancer.

Targeted Therapy	Cellular Model	Therapeutic Alternative
SERM, SERD, AI, CDKI, HERI	Luminal A: HR^+^ HER-2^−^ MCF-7 Luminal B: HR^+^ HER-2^+^ MCF-7/HER Post-menopausal: aromatase positive MCF-7 ^AROM^	Natural phytochemicals nutritional herbs Documented human consumption Lack of detectable systemic toxicity
Systemic toxicity, acquired tumor resistance, drug-resistant stem cell population	Luminal B: HR^+^ HER-2^+^ MCF-7/HER Post-menopausal: aromatase positive MCF-7 ^AROM^	Inhibited proliferation. Increased 2-OHE_1_
	Proliferation: increased by E_2_ Decreased by PRG	
	E_2_ metabolites: proliferative metabolites increased, anti-proliferative metabolites decreased PRG metabolites: proliferative metabolites decreased, anti-proliferative metabolites increased	
	TAM-R stem cells: TS increased CD44, NANOG, and OCT-4 increased	
Future directions: novel pharmacological inhibitors specific for RAS, PI3K, and AKT signaling pathways. Efficacy of small molecule inhibitors on developed stem cell models. Safety and efficacy in Phase 0 clinical trials	Future directions: stem cell models from therapy-resistant PDTX and PDTO. Cellular and molecular characterization of developed stem cell models	Future directions: efficacy of natural phytochemicals and nutritional herbs on PDTX- and PDTO-derived stem cell models.

Overview: Luminal Breast Cancer. This overview summarizes all the aspects that are discussed in the present review. SERM, selective estrogen-receptor modulator; SERD, selective estrogen-receptor degrader; AI, aromatase inhibitor; CDKI, cyclin-dependent kinase inhibitor; HERI, human epidermal growth-factor receptor inhibitor; HR, hormone receptor; HER-2, human epidermal growth-factor receptor-2; HER, human epidermal-growth factor; AROM, aromatase; E_2_, 17β-estradiol; PRG, progesterone; 2-OHE1, 2-hydroxyestrone; TAM-R, tamoxifen resistant; TS, tumor spheroid; CD44; cluster of differentiation 44; NANOG, DNA-binding transcription factor; OCT-4, octamer-binding transcription factor-4; PI3K, phospho-inositidyl-3 kinase; AKT, protein kinase B; PDTX, patient-derived tumor xenograft; and PDTO, patient-derived tumor organoid.

## Data Availability

The data sets used and/or analyzed in the current study are available from the corresponding author on reasonable request.

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
