# Peer review of "The Divergent Effects of Ovarian Steroid Hormones in the MCF-7 Model for Luminal A Breast Cancer: Mechanistic Leads for Therapy"

_ijms, 2022, doi:10.3390/ijms23094800_

Round 1

Reviewer 1 Report

Comments on the manuscript:

“Divergent effects of ovarian steroid hormones in Luminal A breast cancer: Mechanistic leads for therapy”

The effects of the steroid hormones 17β-estradiol and progesterone on endocrine target tissues have long been known. In breast cancer, 17β-estradiol is a growth promoter, while the effects of progesterone are less known. In Luminal A and Luminal B cancers, hormone therapy mainly targets estrogen receptors and their biosynthesis and/or growth factor receptors, but these therapies may be accompanied by undesirable toxic effects such as the emergence of cells drug-resistant cancer strains. New therapies are therefore being sought. This is how progesterone inhibits growth in dose-dependent ways, decreases cell cycle progression and increases apoptosis, and how the adequate use of nutritional plants used in Chinese medicine increases anti-proliferative metabolites. The manuscript is a review of publications dealing with the modulation of growth by hormone metabolites, drug resistant stem cells and nutritional plants in the MCF-7 cell model, with the aim of developing new treatments for Luminal A sub-type of breast cancer.

This article is interesting and useful. It focuses in particular on the effects of plants used in Chinese medicine. The latter, which are the subject of an increasing number of publications, have validated therapeutic effects. This study could be published after some minor corrections.

Title: it would be useful to specify in the title that the study essentially concerns cultures of MCF-7 cells.

Page 3, line 121, table 2: Review the table which does not seem very clear to me: there seems to be a shift in the first lines.

Page 5, line 145, figure 1: “Figure 1” instead of “Figure 1A, B”.

Page 6, line 167, figure 2: “Figure 2” instead of “Figure 2A, B”.

Page 8, lines 254-255: use italics to write Epimedium grandiflorum and Lycium barbarum. It is Lycium barbarum, not Lycium barbarun

Page 9, line 278: use italics to write the name of genes.

Page 9, line 289: (9, 54, 55]: use [ instead of (.

Page 11, line 341: delete “progesterone” after PRG.

Author Response

The following is my point-by-point response to the reviewer comments. The recommended changes are included in bold face for the information and approval of the editor and the referees.  

Reviewer # 1:

  1. Concern: The title should be modified to include MCF-7 cells.

Response: The title is modified as follows.

“Divergent effects of ovarian steroid hormones in the MCF-7 model for Luminal A breast cancer: Mechanistic leads for therapy”.

  1. Concern: page 3, line121, Table 2: resolve the problems regarding alignment of the content.

Response: The Table has been reformatted to correct the alignment of the content.

  1. Concern: page 5, line 145, figure 1, instead of Figure 1A, B.

Response: The figure 1A, B has been corrected as Figure 1.

  1. Concern: page 6, line 167, Figure 2, instead of Figure 2A, B.

Response: The Figure 2A, B has been correct as Figure 2.

  1. Concern: page 8, lines 254-255: Use italics for Epimedium grandiflorum and Lycium barbarum.

Response: The botanical names of the herbs are provided in italic font.

  1. Concern: page 9, line 278: Use italics for the genes.

Response: The names of the genes are provided in italic font.

  1. Concern: Page 9, line 289: Use square brackets instead of parenthesis for the references numbers.

Response: The parentheses have been changed to square brackets.

  1. Concern: page 11, line 341: delete “progesterone” after PRG.

Response: “progesterone” is not deleted because it represents the expansion of the abbreviation PRG.  

Reviewer 2 Report

Dear Author,

I congratulate you on the interesting topic. Your paper could be a stimulus towards a deeper evaluation of the argument provided. The paper is a well written overview of the actual situation, providing a proof of concept for further applications. Really interesting is the possibility to use non-toxic nutritional herbs with their inhibitory efficacy on cell proliferation, especially in consideration of the drug resistance that could compromises therapeutic efficacy.

Given these considerations, after a minor spell check, I think that your paper could be accepted for publication present form.

Kind Regards

Author Response

Reviewer # 2:

  1. Concern: Do a spell check of the manuscript.

Response: A spell check has been done for the entire manuscript.